# Molecular and Morphological Assessment of *Septoria* Species Associated with Ornamental Plants in Yunnan Province, China

**DOI:** 10.3390/jof7060483

**Published:** 2021-06-16

**Authors:** Yuan-Yan An, Monika C. Dayarathne, Xiang-Yu Zeng, Alan J. L. Phillips, Kevin D. Hyde, Yong Wang

**Affiliations:** 1Department of Plant Pathology, College of Agriculture, Guizhou University, Guiyang 550025, China; anyuanyan123@sina.com (Y.-Y.A.); monidaya40@gmail.com (M.C.D.); jason.xyzeng@gmail.com (X.-Y.Z.); 2Center of Excellence in Fungal Research, Mae Fah Luang University, Chiang Rai 57100, Thailand; kdhyde3@gmail.com; 3Faculdade de Ciências, Biosystems and Integrative Sciences Institute (BioISI), Universidade de Lisboa, Campo Grande, 1749-016 Lisbon, Portugal; alan.jl.phillips@gmail.com; 4Institute of Plant Health, Zhongkai, University of Agriculture and Engineering, Haizhu District, Guangzhou 510225, China

**Keywords:** GCPSR, molecular assessment, new taxa, *Septoria*

## Abstract

The Karst landform is the main geographic characteristic in South China. Such areas are rich in vegetation and especially suitable for growth of shrubs and herbaceous plants. In this study, 11 *Septoria* strains were obtained from different plants’ leaves collected in the Kunming Botanical Garden, Yunnan Province, China. Based on single-gene and multi-gene analyses of five gene loci (*tef1*, *rpb2*, *tub2*, ITS, and *LSU*) and four gene regions (without *LSU*), these strains were found to belong to three independent phylogenetic lineages representing five species, including four novel taxa, and one new record for China. Five single gene trees were also provided to evaluate the effectiveness of each gene for discriminating the species, as a result of which *tub2* was found to have the most suitable DNA barcode for rapid identification. Morphological descriptions, illustrations, and comparisons are provided for a more comprehensive assessment. Genealogical Concordance Phylogenetic Species Recognition (GCPSR) with a pairwise homoplasy index (PHI) test was used to evaluate the conclusions of the phylogenetic analyses.

## 1. Introduction

*Septoria* Sacc., established by Saccardo in 1884, belongs to the Mycosphaerellaceae family of fungi and accommodates around 1000 species [1,2], although only 200 species have been confirmed by molecular data [2]. Many of these species cause leaf spot diseases of numerous cultivated and wild plants [3]. According to its morphology at the primary generic level, *Septoria* includes coelomycetous asexual morphs, which produce pycnidial conidiomata having holoblastic, hyaline, smooth, filiform-to-cylindrical multi-septate conidia [4,5,6,7,8,9]. On the basis of a polyphasic approach to taxon delimitation, Verkley et al. [3] pointed out that septoria-like fungi preserved in CBS were in fact distributed over three main clades and introduced a novel genus: *Caryophylloseptoria* Verkley, Quaedvlieg and Crous. Quaedvlieg et al. [10] re-defined *Septoria* as having pycnidial to acervular conidiomata and hyaline conidiophores that give rise to conidiogenous cells that proliferate both sympodially and percurrently to form hyaline, filiform conidia with transverse eusepta. Crous et al. [11] introduced *Acervuloseptoria* on account of its black, erumpent conidiomata, and the old name *Septoria capensis* G. Winter was transferred to this genus [12]. More DNA sequence data are necessary to support the morphological characters in this species identification [10].

In this study, 11 *Septoria* strains were obtained from different ornamental plants in a South China Karst region. Morphological comparisons, phylogenetic analyses based on five gene loci, DNA base-pair differences, and GCPSR evaluation confirmed that they formed three phylogenetic lineages representing five *Septoria* species comprising four novel species and one new Chinese record.

## 2. Materials and Methods

### 2.1. Fungus Collection and Isolation

The isolates included in this study were collected from the Kunming Botanical Garden, Yunnan Province, China, in 2018. Pure cultures were obtained by single-spore isolations following the methods of surface sterilization and incubation of specimens [13]. After 24 h of incubation, germinated conidia were transferred to the new potato-dextrose agar (PDA) medium and incubated at 25 °C. The holotype specimens were deposited in the Herbarium of the Department of Plant Pathology, Agricultural College, Guizhou University (HGUP). The type cultures were deposited in the Culture Collection at the Department of Plant Pathology, Agriculture College, Guizhou University, China (GUCC), and the Mae Fah Luang University Culture Collection (MFLUCC) in Thailand (Table 1).

### 2.2. Morphological Studies

Morphological characters were recorded from cultures that had been incubated for 2 to 3 weeks. For light microscopy, the relevant structures were mounted in Shear’s liquid, distilled water or lactic acid and examined with an Olympus BX53 microscope. Measurements of 30 conidia and other structures were made at a magnification of 1000× [14]. Taxonomic information of the new taxa was submitted to the MycoBank database (www.mycobank.org, accessed on 24 March 2021).

### 2.3. DNA Extraction, Amplification (PCR), and Sequencing

Methods outlined in [15] were followed for DNA extraction, amplification (PCR), sequencing, and phylogenetic analysis. Fresh fungal mycelia of strains were harvested using a sterile scalpel, and the genomic DNA was isolated using A BIOMIGA Fungus Genomic DNA Extraction Kit (GD2416) according to the manufacturer’s protocol. The DNA was amplified in a 25 μL reaction volume containing 2.5 μL 10× PCR buffer, 1 μL of each primer (10 μM), 1 μL template DNA, 0.25 μL Taq DNA polymerase (Promega, Madison, WI, USA), and 18.5 μL ddH_2_O. Five gene regions—loci β-tubulin (*tub2*), internal transcribed spacer (ITS), Translation elongation factor 1-alpha (*tef1*), 28S nrDNA (*LSU*), and RNA polymerase II second largest subunit (*rpb2*)—were targeted for Polymerase Chain Reaction (PCR) amplification and subsequent sequencing. The primers used and amplification conditions of the genes are listed in Table 2. The DNA sequences were submitted to GenBank and their accession numbers are provided in Table 1. The generated sequences for each locus and the reference sequences of ex-type or representative sequences of *Septoria* species downloaded from GenBank (Table 1) were aligned with the online version of MAFFT v. 7.307 [16,17].

### 2.4. Phylogenetic Analyses 

The alignments were checked and manually improved where necessary using MEGA v. 5 [27]. Phylogenetic analyses were performed by maximum parsimony (MP), maximum likelihood (ML), and Bayesian methods for individual and combined locus datasets. Ambiguous regions were excluded from the analyses and gaps were treated as missing data. Maximum parsimony analysis was performed in PAUP v. 4.0b10 [28] using the heuristic search option with 100 random taxon additions and tree bisection and re-connection (TBR) as the branch-swapping algorithm with Maxtrees = 5000. Branches of zero length were collapsed and all multiple, and equally most parsimonious trees were saved. The robustness of the trees obtained was evaluated by 1000 bootstrap replications [29]. Other measures calculated included tree length (TL), consistency index (CI), retention index (RI), and rescaled consistency index (RC).

The resulting PHYLIP file was used to generate the ML tree on the CIPRES Science Gateway [30] using RAxML-HPC2 black box with 1000 bootstrap replicates and GTRGAMMA as the nucleotide substitution model. Bayesian analyses were launched with random starting trees for 10,000,000 generations. The heat parameter was set at 0.15 and trees were saved every 1000 generations until the average standard deviation of split frequencies reached 0.01 (stop value). Burn-in was set to 25% after which the likelihood values were considered to be stationary. All resulting trees were visualized with FigTree v. 1.4.3 (Institute of Evolutionary Biology, University of Edinburgh, UK) [31].

### 2.5. Genealogical Concordance Phylogenetic Species Recognition Analysis

The Genealogical Concordance Phylogenetic Species Recognition (GCPSR) concept with a pairwise homoplasy index (PHI) test was used to analyze the new species, their species boundaries, and their most closely related taxa as described by Quaedvlieg et al. [32]. The recombination level within phylogenetically closely related species was determined with the PHI test performed using SplitsTree4 [33,34]. The concatenated datasets (*tef1*, *rpb2*, *tub2*, ITS, and *LSU*) were used. The relationships between different taxa were visualized in splits graphs with both the Log-Det transformation and splits decomposition options. A pairwise homoplasy index below a 0.05 threshold (Fw < 0.05) indicated the presence of significant recombination in the dataset.

## 3. Results

### 3.1. Phylogenetic Analyses

Eleven *Septoria* strains isolated from different plant hosts were sequenced. PCR products of 450–536 bp (*tef1*), 440–453 bp (*tub2*), 458–524 bp (ITS), 799–863 bp (*LSU*), and 718–1083 bp (*rpb2*) were obtained. By alignment with the single-gene region and then in combination in the order of *tef1*, *rpb2*, *tub2*, ITS, and *LSU* with *Cercospora beticola* (CBS 124.31), 2434 characters were obtained: *tef1*, 1–479; *rpb2*, 480–824; *tub2*, 825–1149; ITS, 1159–163; and *LSU*, 1636–2434. Among these characters, 1672 were constant, while 195 variable characters were parsimony-uninformative and 567 were parsimony informative. The parameters of the MP phylogenetic trees are shown in Table 3, and the procedure yielded a single most parsimonious tree (Figure 1). Similar topologies were obtained by MP, ML, and Bayesian methods. In the *Septoria* phylogenetic tree (Figure 1), all *Septoria* isolates were grouped together, but only the BI support was high (BPP = 1), while the three major clades received greater statistical support (Branch 1: ML/BI = 98%/0.99; Branches 2: MP/ML/BI = 88%/87%/0.99; Branch 3: MP/ML/BI = 88%/80%/1.00). Six strains (GUCC 2131.1, GUCC 2131.2, GUCC 2131.3, GUCC 2131.4, GUCC 2164.1, and GUCC 2164.2) were grouped in the clade that included *S. posoniensis* and *S. exotica* (MP: 95%, ML: 92% and BPP: 0.94) in Branch 1. In this group, five strains (GUCC 2131.2, GUCC 2131.3, GUCC 2131.4, GUCC 2164.1 and GUCC 2164.2) formed an independent branch adjacent to GUCC 2131.1 and *S. posoniensis* (MP: 76%, ML: 86%, and BPP: 0.95), but these five strains were split into two sub-branches: one containing GUCC 2131.2, GUCC2164.1, and GUCC2164.2, and the other containing GUCC 2131.3 and GUCC2131.4, with good support (MP: 75%; BPP: 1.00). Strain GUCC 2127.3 was aligned to the branch that included *S. chamaecisti, S. citri, S. citricola, S. protearum,* and *S. limonum* with high statistical support (MP: 98%, ML: 100% and BPP: 1) but small phylogenetic distances. Strains GUCC 2164.3, GUCC 2164.4, GUCC 2127.1, and GUCC 2127.4 formed a strongly supported group (MP: 95%; ML: 100%; BPP: 1.00) closely related to *S. coprosmae* and *S. verbenae* with good support values (MP: 85%; BPP: 0.96). In Branch 2, four strains clustered in a clade in which GUCC 2127.1, GUCC2164.3 and GUCC2164.4 formed a sub-group, were very close to GUCC 2127.4, supported by high statistical values (MP: 95%, ML: 100%, and BPP: 1).

We also compared the DNA base-pair differences in five different loci between our strains and related species (Appendix A). This revealed that the *LSU* gene region was too conserved for species-level identification, and the ITS had little value, but *tef1*, *tub2*, and *rpb2* provided more than 80% of the DNA base-pair differences (Appendix A). We also built a phylogenetic tree based on four loci, excluding the *LSU* region (Figure 2), using the parameters for MP analysis in Table 3. The topology showed highly similar placements of our strains in the *Septoria* in Figure 1; however, in Figure 2 only two branches were formed and all members of Branch 3 were integrated with Branch 1. To evaluate the distinctive effectiveness of different DNA markers, five single gene trees were constructed (Appendix A) and all MP parameters were as indicated in Table 3. Through comparison, we found that only *tub2* and *tef1* included more parsimonious characters (50.7% and 49.2%), and the sequence of *tub2* was shorter than that of *tef1*. Moreover, the topology originating from the *tub2* gene region was more similar to Figure 2.

### 3.2. Genealogical Concordance Phylogenetic Species Recognition

In order to determine evolutionary independence, the GCPSR concept was applied to the GUCC 2164.2, GUCC 2131.4, GUCC 2131.1, and related taxa *S. chrysanthemella* (CBS 128716), *S. exotica* (CBS 163.78), and *S. posoniensis* (CBS 128645). A pairwise homoplasy index (PHI or Fw) less than 0.05 provided evidence of the presence of significant recombination within a dataset. According to the GCPSR analysis, our dataset showed PHI of 0.116, indicating no significant genetic recombination among our strains and related taxa. Hence, it was concluded that these taxa were significantly different from each other.

For GUCC 2164.3 and GUCC 2127.4 and related species *S. coprosmae* (CBS 113391) and *S. verbenae* (CBS 113438), the pairwise homoplasy index (PHI or Fw) was 1.173 × 10^−8^, which provided evidence for the presence of significant recombination within a dataset. The four strains could belong to a single species.

### 3.3. Taxonomy

(1)Septoria sanguisorbigena Y.Y. An & Yong Wang bis, sp. nov. (Figure 3)

MycoBank MB 839125

Etymology: The name refers to the plant host, from which the fungus was collected.

Description in vitro: *Colonies*: on PDA 15–25 mm diameter after 2 weeks with a undulating even margin, restricted, irregularly pustulate; the surface almost black with low and finely felted diffuse, grey-to-white aerial mycelium. *Conidiomata*: pycnidial, epiphyllous, immersed, subglobose to globose, black, 120–250 µm diameter; ostiolum central, circular, initially 25–35 µm wide, later becoming more irregular and up to 100 µm wide, conidiomatal wall 20–40 µm thick, composed of an outer layer of angular-to-irregular cells mostly 4.5–10 µm diameter with pale to orange-brown walls and an inner layer of isodiametric, hyaline cells 7–20 μm diamater. *Conidiogenous cells*: hyaline, discrete, holoblastic, sympodially or percurrently proliferating, ampulliform, 4.5–8 × 1.5–2.5 µm (avg. = 5.6 × 2 µm, *n* = 30). *Conidia*: hyaline, filiform, straight to somewhat flexuous, the upper cell tapered into obtuse apex, relatively wide truncated base, (1–)3–5(–7) septate, not or only indistinctly constricted at the septa, contents granular or with minute oil-droplets around the septa and at the ends, 12.5–30 × 0.6–2 µm (avg. = 20.5 × 1.3 µm, *n* = 30). Sexual morph unknown.

Type: CHINA, Yunnan Province, Kunming Botanical Garden, from leaves of *Sanguisorba officinalis* L., February 2018, Y.Y. An (HGUP 2164.2, holotype); ex-type culture GUCC 2164.2; isotype culture MFLUCC 20-0185.

Other material examined: CHINA, Yunnan Province, Kunming Botanical Garden, from leaves of *Sanguisorba officinalis*, February 2018, Y.Y. An (HGUP 2164.2); from leaves of *Pilea cadierei* Gagnep. & Guillaumin, February 2018, Y.Y. An (HGUP 2131.2).

Notes: Phylogenetic analyses confirmed that three strains (GUCC 2131.2, GUCC 2164.1, and GUCC 2164.2) had a close relationship with *S. chrysanthemella*, *S. exotica*, *S.*
*longipes*, *S. pileicola,* and *S. posoniensis* and this was supported by credible statistic values of the MP and ML methods (Figure 1). However, the independent branch only included those strains with high support values (MP: 95%, ML: 90%, and BPP: 0.99) adjacent to *S. pileicola* with moderate MP bootstrap but 1.00 BPP support. The new species had narrower conidia (0.6–2 µm) with 3–5 septa than those of *S. pileicola* (1.5–3.5 µm) with only 1–2 septa. In addition, this new taxon had obviously smaller conidia (12.5–30 × 0.6–2 µm) than *S. chrysanthemella* (34–66 × 2.5–3 µm) and *S. longipes* (17–46.5 × 1.5–2.5 µm) [35]. *Septoria posoniensis* and *S. exotica* have longer conidia, which was different to the new species [33,34]. DNA base differences indicated these three strains had nearly the identical sequence data (only two different bases on ITS region), but on protein-coding genes possessed more differences to distinguish them from related species (Appendix A). GCPSR test also provided a powerful proof to clarify them as different species.

(2)Septoria pileicola Y.Y. An & Yong Wang bis sp. nov. (Figure 4)

MycoBank MB 839126

Etymology: The name refers to the plant host from which the fungus was collected.

Description in vitro: *Colonies*: on PDA up to 10–15 mm diameter, with an even, glabrous, colourless margin in 2 weeks. *Mycelium*: greenish grey to dark slate-blue, immersed, throughout covered by well-developed, tufty whitish-grey aerial mycelium that later attains a reddish haze; reverse black, but margin paler; in the central part of the colony numerous pycnidia develop, releasing pale vinaceous to rosy-buff conidial ball. *Conidiomata*: pycnidial, epiphyllous but sometimes also visible from the underside of the lesion, one to a few in each leaf spot, subglobose to globose, brown to black, usually fully immersed, 80–120 µm diam. *Ostiolum*: central, initially circular and 15–30 µm wide, later becoming more irregular and up to 45 µm wide, surrounding cells concolorous to pale brown. *Conidiogenous cells*: hyaline, discrete, doliiform, or narrowly to broadly ampulliform, holoblastic, with a relatively narrow elongated neck, proliferating percurrently several times with distinct annellations, often also sympodially after a few percurrent proliferations, 5.5–12 × 2–3.5 µm. *Conidia*: cylindrical or filiform-cylindrical, straight to slightly curved, narrowly to broadly rounded at the apex, narrowing slightly or more distinctly to a truncate base, (0–)1–2-septate, not or slightly constricted around the septa, hyaline, contents with a few minute oil-droplets and granular material in each cell in the rehydrated state, 8.5–30 × 1.5–3.5 µm. Sexual morph unknown.

Type: CHINA, Yunnan Province, Kunming Botanical Garden, from leaves of *Pilea cadierei* Gagnep. & Guillaumin, February 2018, Y.Y. An (HGUP 2131.4, holotype); ex-type culture GUCC 2131.4; isotype culture MFLUCC 20-0184.

Other material examined: CHINA, Yunnan Province, Kunming Botanical Garden, from leaves of *Pilea cadierei*, February 2018, Y.Y. An (HGUP 2131.3).

Note: Phylogenetic analyses based on five gene regions showed that *Septoria pileicola* strains GUCC 2131.3 and GUCC 2131.4 are closely related to *S. chrysanthemella*, *S. exotica*, *S. longipes*, *S. posoniensis,* and *S.*
*sanguisorbigena* (Figure 1), but formed a subclade with *S. sanguisorbigena*. After morphological comparisons, we found that *Septoria pileicola* can be distinguished from *S. sanguisorbigena* by its wider conidia, and from *S. posoniensis* and *S. exotica* by its shorter conidia with obviously fewer septa [36,37]. For *S. chrysanthemella* and *S. longipes*, the species had apparently shorter conidia [35]. The two strains of *Septoria pileicola* had nearly the identical sequences (only one different ITS base pair); however, the *tub2* gene provided enough base distinction to separate it from related species (Appendix A) according to the guidelines of Jeewon and Hyde [38]. The PHI value was 0.116 (>0.05), indicating no significant genetic recombination among *S. pileicola*, *S. sanguisorbigena*, *S. chrysanthemella, S. exotica,* and *S. posoniensis*. Thus, they should belong to different species [39].

(3)Septoria longipes Y.Y. An & Yong Wang bis sp. nov. (Figure 5)

MycoBank MB 839127

Etymology: The name refers to the long conidia of this species.

Description in vitro: *Colonies*: on PDA 11–15 mm diameter, with an even, light brown to dark-brown margin in 2 weeks; immersed mycelium grey to dark slate-blue in the center, black near the margin. *Aerial mycelium*: well-developed, white to snow white, covering the colony surface. *Conidiomata*: pycnidial, numerous, mostly epiphyllous, semi-immersed, black, mostly 80–200 µm diameter, with a central, first narrow, later wider opening, releasing pale white cirrhi of conidia. *Conidiomatal wall*: one or two layers of brown-walled, angular cells, lined by a layer of hyaline cells. *Conidiogenous cells*: hyaline, discrete, holoblastic, sympodially or percurrently proliferating, ampulliform, 8–16 × 1.5–5.5 µm. *Conidia*: filiform to filiform-cylindrical, straight, flexuous or curved, attenuated gradually to the narrowly rounded to pointed apex, attenuated gradually or more abruptly to the narrowly truncate base, (0–)3–5(–8)-septate, 17–46.5 × 1.5–2.5 µm. Sexual morph unknown.

Type: CHINA, Yunnan Province, Kunming Botanical Garden, from leaves of *Pilea cadierei* Gagnep. & Guillaumin, February 2018, Y.Y. An (HGUP 2131.1, holotype); ex-type culture GUCC 2131.1)

Notes: Only one strain (GUCC 2131.1) of this taxon was available. It clustered with *S. posoniensis* supported by MP (70%) and Bayesian (0.93) analyses and is closely related to *S. chrysanthemella*, *S. exotica*, *S. pileicola,* and *S.*
*sanguisorbigena*. Morphological comparisons indicated that GUCC 2131.1 differed from *S. posoniensis* by conidia by more septa, and from *S. chrysanthemella* (4–10 × 5–6 µm) by larger conidiogenous cells (8–16 × 1.5–5.5 µm) [35,36]. This species produced longer conidia than *S. pileicola* and *S.*
*sanguisorbigena*. It was confirmed that two protein-coding genes, except for *tef1*, provided enough base distinction with related species (Appendix A). GCPSR test also supported them as different species.

(4)Septoria dispori Y.Y. An & Yong Wang bis sp. nov. (Figure 6)

MycoBank MB 839128

Etymology: The name refers to the plant host from which the fungus was collected.

Description in vitro: *Colonies*: on PDA 2.0–3.5 mm diameter, with an even to slightly ruffled, glabrous, dull yellow margin in 2 weeks, spreading, remaining almost plane, immersed mycelium yellowish brown to brown; aerial mycelium well-developed, goose feather flocculent on the surface of the colony; numerous conidiomatal initials developing at the surface, mature ones releasing cirrhi of conidia that first are milky white, later salmon, sometimes merging to form slimy masses covering areas of the colony surface. *Conidiogenous cells*: hyaline, broadly or elongated ampulliform, normally with a distinct neck, hyaline, holoblastic, proliferating percurrently, annellations indistinct, 10–15 ×1.5–2.5 µm. *Conidia*: cylindrical to filiform-cylindrical, slightly to strongly curved, rarely somewhat flexuous, narrowly rounded to pointed at the apex, attenuated gradually or more abruptly towards a narrowly truncate base, 3–5–8-septate, later with secondary septa dividing the cells, sometimes breaking up into smaller fragments in the cirrhus, not or slightly constricted around the septa, hyaline, 14–41.5 × 1.5–2.5 µm. Sexual morph unknown.

Type: CHINA, Yunnan Province, Botanical Garden of Kunming country, from leaves of *Disporum bodinieri* (Levl. et Vaniot.) Wang et Y. C. Tang, February 2018, Y.Y. An (HGUP 2127.1, holotype); ex-type culture GUCC 2127.1.

Other material examined: CHINA, Yunnan Province, Kunming Botanical Garden, from leaves of *Disporum bodinieri*, February 2018, Y.Y. An (HGUP 2127.4); from leaves of *Sanguisorba officinalis* L., February 2018, Y.Y. An (HGUP 2164.3 and HGUP 2164.4).

Note: Four strains (GUCC 2127.1, GUCC 2127.4, GUCC 2164.3, and GUCC 2164.4) of *Septoria dispori* clustered together with high statistical support (MP: 95%, ML: 100%, BPP: 1.00) adjacent to *S. coprosmae* and *S. verbenae*. Thus, we consider these four strains to be a single species. *Septoria coprosmae* produced spermatogonia of an *Asteromella*-state, but this species did not [40]. Conidia of *S. verbenae* possessed fewer septa than those of *Septoria dispori* [41]. GUCC 2127.4 showed some phylogenetic distance from the other three strains, however DNA base comparison (Appendix A) revealed only 11 bases that had *tub2* differences. The PHT test confirmed significant recombination between strains GUCC 2164. 4 and GUCC 2164.3 and they were morphologically similar. Thus by combining the above evidence, we established the four strains as a new taxon.

(5)Septoria protearum Viljoen & Crous, in Swart, Crous, Denman & Palm, S. Afr. J. Bot. 64(2): 144 (1998) (Figure 7)

Description in vitro: *Colonies*: on PDA 15–25 mm with an even, glabrous white margin in 2 weeks, plane spreading, immersed. *Mycelium*: pink, lacking aerial hypha. *Conidiomata* developing after 1 week, mostly immersed and releasing whitish conidial slime. *Conidiogenous cells*: hyaline, cylindrical, broadly to narrowly ampulliform, with a distinct neck of variable length, holoblastic, with several distinct percurrent proliferations, more rarely also sympodial after a sequence of percurrent proliferations of the same cell, 5–10(–13.5) × 2.5–3(–3.5) µm. *Conidia*: filiform, straight, more often irregularly curved, 0–4 septate, not or only inconspicuously constricted around the septa, hyaline, 16–25 × 2.5–3.5 µm. Sexual morph unknown.

Material examined: CHINA, Yunnan Province, Kunming Botanical Garden, from leaves of *Disporum bodinieri* (Levl. et Vaniot.) Wang et Y. C. Tang, February 2018, Y.Y. An (HGUP 2127.3), living culture GUCC 2127.3.

Notes: DNA base comparison (Appendix A), revealed that sequences of strain GUCC 2127.3 were identical to the ex-type strain of *S. protearum* (CBS 778.97) in four gene regions. Conidial shape and size range of *S. protearum* (12–22 × 1.5–2 µm) were similar to the present strain [42]. The number of conidial septa of the two strains was also the same (0–4 septa). Thus, we conclude that GUCC 2127.3 is *S. protearum.*

## 4. Discussion

Verkley et al. [3] pointed out that for the identification of the *Septoria* species, morphological description must be integrated with sequences analyses. Quaedvlieg et al. [10] treated species in *Septoria* within a modern taxonomic framework and pointed out that *Septoria* spp. formed a well-defined phylogenetic clade. Regarding morphology, the species concept was to produce pycnidial, ostiolate conidiomata; conidiophores reduced to conidiogenous cells that proliferate sympodially; and hyaline, filiform conidia with transverse eusepta that fit the original concept of [4]. We followed this system and applied morphological and phylogenetic approaches to the present study. After comparing the topologies of five single-gene and two multi-gene trees (Figure 1 and Figure 2 and Appendix A), we showed that *Septoria* forms two branches (Branch 1 and Branch 2), mainly because only the phylogenetic trees based on the *LSU* region and five DNA fragments (including *LSU*) supported three branches, whereas the conserved *LSU* sequences included the least parsimonious characters (31/799) (Table 3). In morphology, all species in Branch 3 produced filiform or fusiform, sub-straight to slightly curved conidia mainly with 3 septa, which was not a unique characteristic. Thus, we proposed exclusion of the *LSU* region for multi-gene analyses of *Septoria* at the species level, but always as the primary DNA barcode with more parsimonious characters (43/486), the ITS fragment was conserved in the present phylogenetic analysis.

The *S. protearum* complex accommodated eight members: *S. citri*, *S. citicola*, *S. chamaecisti, S. gerberae*, *S. hederae*, *S. lobelia*, *S. limonum*, and *S. protearum*, according to Verkley et al. [3]. Apart from *S. protearum*, the other species were old names without ex-type cultures, and thus no sequences were available. The base comparison of DNA sequences originated from Verkley et al. [3], who indicated that among these eight species there were only approximately 10 base-pair differences on the *rpb2* fragment (434 characters) of *S. gerberae*, *S. hederae,* and *S. lobelia* compared to the other five species, while for the other four gene regions, their sequences were nearly identical (≤1 base difference) (Appendix A). On the other hand, in the literature these seven species are depicted only by simple descriptions often without drawings or photographs, which does not strongly support them as different taxa. Comparing with the sequences from Verkley et al. [3] and in the absence of type materials, we were more willing to believe that they belonged to the same species, *S. protearum*.

Our 11 strains isolated from *Disporum bodinieri*, *Pilea cadierei*, *Sanguisorba officinalis* all from the Botanical Garden of Kunming county represented five *Septoria* species and included four novel species supported by morphology and phylogeny. *Septoria sanguisorbigena* was obtained from two plant hosts (*Sanguisorba officinalis* and *P. cadierei*), and *S. dispori* was also on two hosts (*D. bodinieri* and *Sanguisorba officinalis*). *Septoria pileicola* and *S. longipes* were only discovered on one host (*P. cadierei*). Our *S. protearum* strain was on *D. bodinieri*. Verkley et al. [3] recalled that trans-family host jumping must be a major force driving the evolution of *Septoria*. Our results support this hypothesis as we found the same species on different hosts. However, our findings revealed that the *Septoria* species did not show any host specialization, which differs from the view of Verkley et al. [3].

## 5. Conclusions

In this study, our 11 *Septoria* strains represented five species including four novel taxa, and one new record for China by morphological comparison and multi-gene analyses. The *Septoria* species are pathogens often causing leaf spot diseases of many plant hosts worldwide [10]. Based on previous studies, relatively sufficient reference sequences are available for rapid identification of *Septoria* pathogens. By comparing the parsimonious-informative characters of different DNA fragments (Appendix A), we showed that either *tef1* or *tub2* is suitable as a secondary DNA barcode, and that the latter was more discriminating than the former. Moreover, the DNA fragment of *tub2* (≈300 bp) was shorter than that of *tef1* (≈450 bp) with a high PCR amplification success rate. Consequently, a standardized approach including morphological characters and phylogenetic analysis is needed for the correct and precise identification of *Septoria* isolates.

## Figures and Tables

**Figure 1 jof-07-00483-f001:**
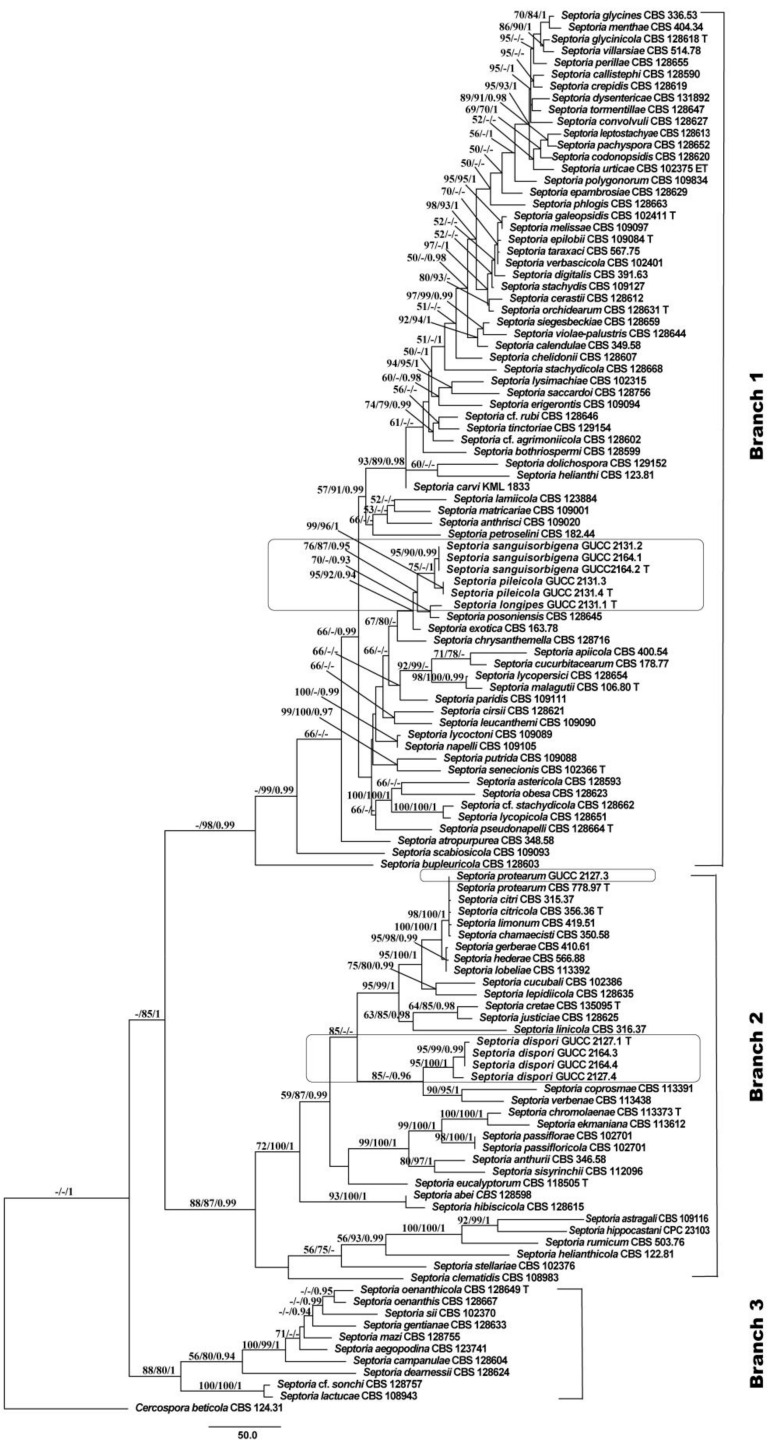
Maximum Parsimony (MP) topology of *Septoria* generated from a combination of *tef1*, *rpb2*, *tub2*, ITS, and *LSU* sequences. *Cercospora beticola* (CBS 124.31) was used as outgroup taxon. MP and ML above 50% and BPP above 0.90 were placed close to topological nodes and separated by “/”, otherwise were labeled with “-”.

**Figure 2 jof-07-00483-f002:**
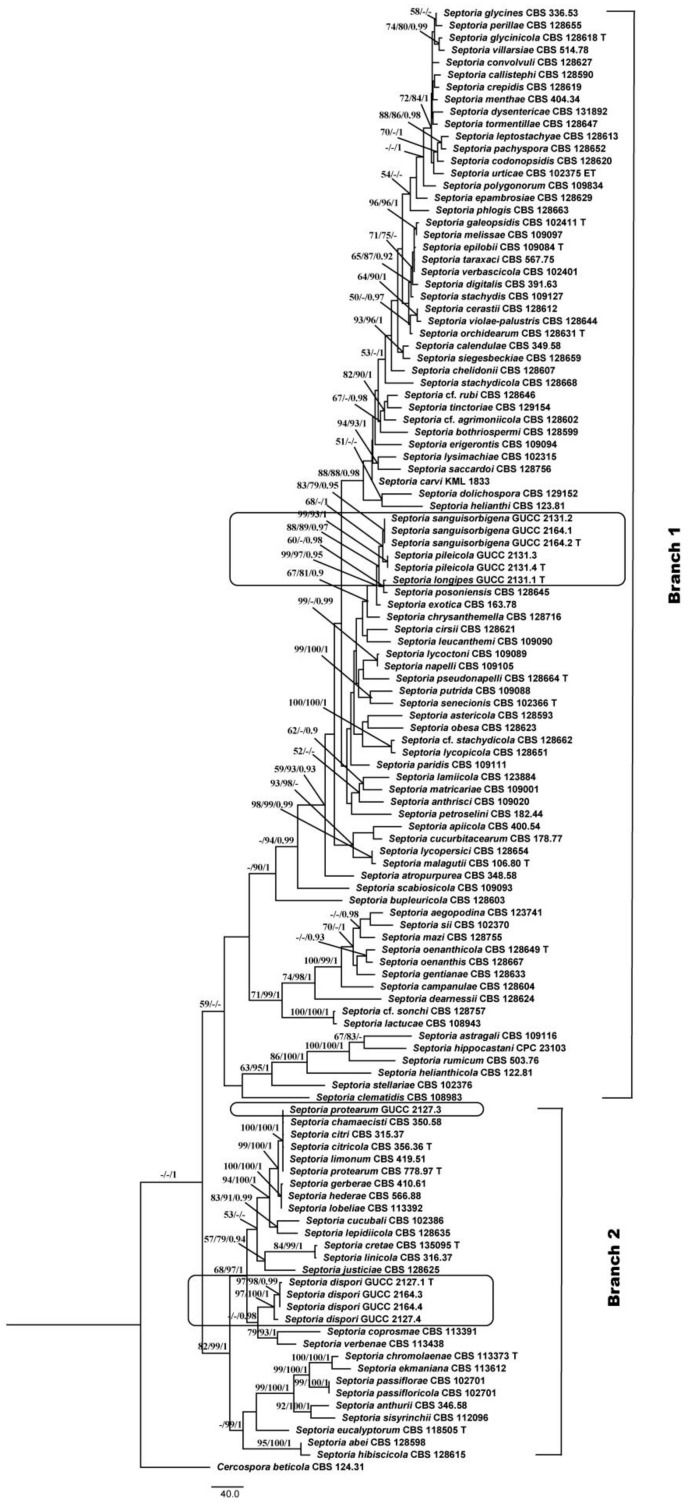
Maximum parsimony (MP) topology of *Septoria* generated from a combination of *tef1*, *rpb2*, *tub2*, and ITS sequences. *Cercospora beticola* (CBS 124.31) was used as an outgroup taxon. MP and ML above 50% and BPP above 0.90 were placed close to topological nodes and separated by “/”; otherwise, they were labeled with “-”.

**Figure 3 jof-07-00483-f003:**
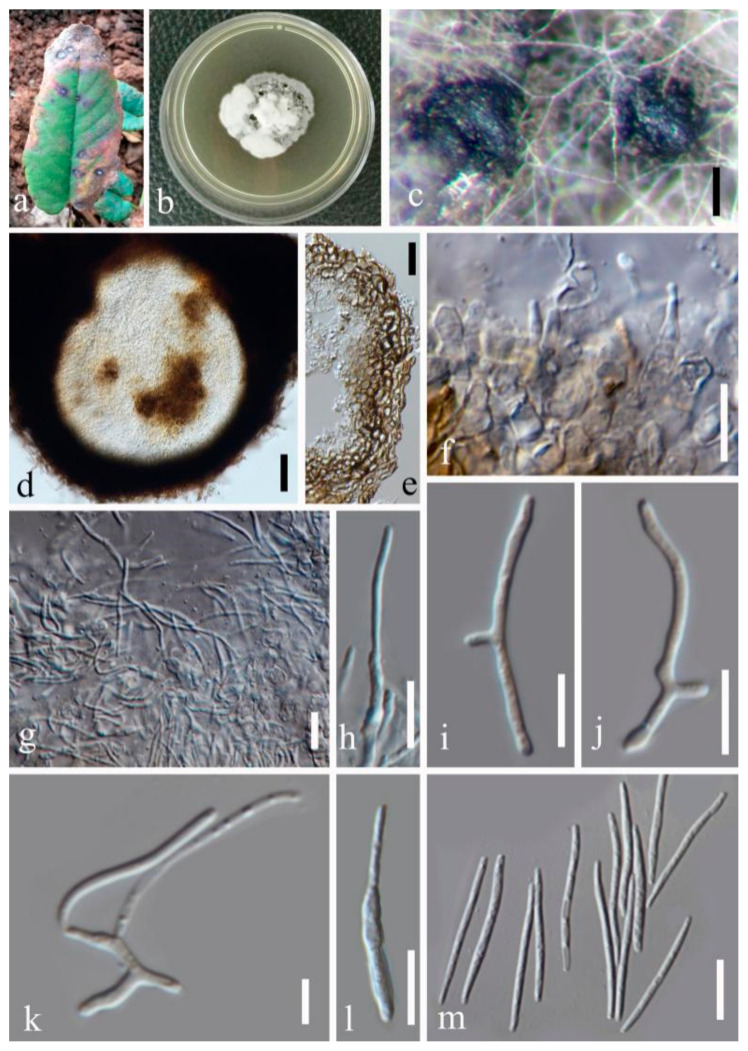
*Septoria sanguisorbigena* (GUCC 2164.2) (**a**) Leaf spot symptoms on the host. (**b**) Colony on PDA culture. (**c**) Conidiomata formed on PDA culture. (**d**) Section through a conidioma. (**e**) Peridium. (**f**–**k**) Conidiogenous cells, (**l**,**m**) Conidia. Scale bars: (**c**) = 25 µm, (**d**) = 50 µm, (**e**) = 20 µm, (**f**–**j**) = 10 µm, (**k**) = 5 µm, (**l**,**m**) = 10 µm.

**Figure 4 jof-07-00483-f004:**
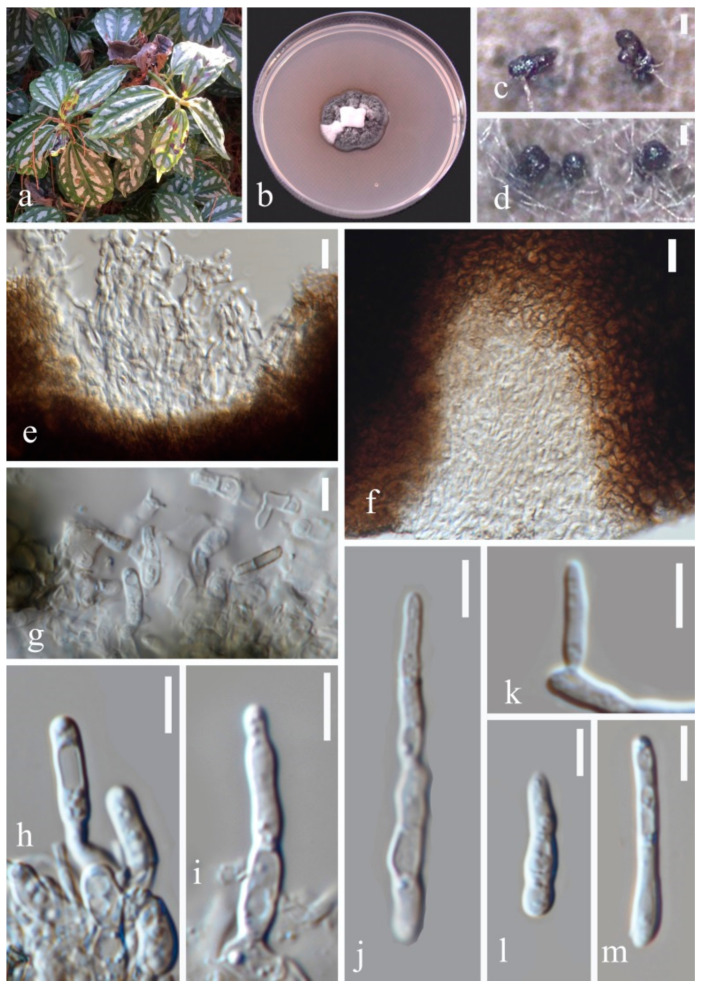
*Septoria pileicola* (GUCC 2131.4): (**a**) Leaf spot symptoms on the host. (**b**) Colonies on PDA culture. (**c,d**) Conidiomata on PDA culture. (**e,f**) Section though conidioma. (**g**–**k**) Conidiogenous cells. (**l,m**) conidia. Scale bars: (**c**,**d**) = 125 µm, (**e**–**g**) = 10 µm, (**h**–**m**) = 10 µm.

**Figure 5 jof-07-00483-f005:**
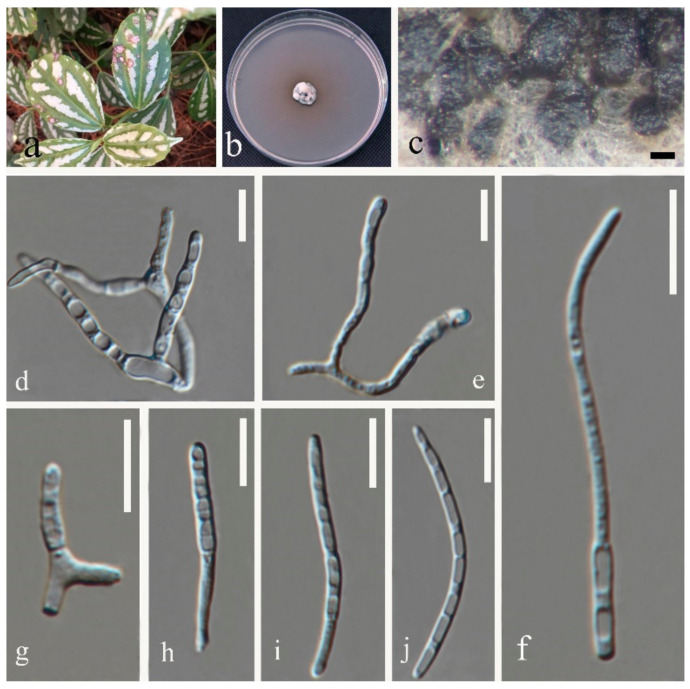
*Septoria longipes* (GUCC 2131.1) (**a**) Leaf spot symptoms on the host. (**b**) Colony on PDA. (**c**) Conidiomata on PDA culture. (**d**–**g**) Conidiophores, Conidiogenous cells and conidia. (**h**–**j**) Conidia. Scale bars: (**c**) = 20 µm. (**d**–**j**) = 10 µm.

**Figure 6 jof-07-00483-f006:**
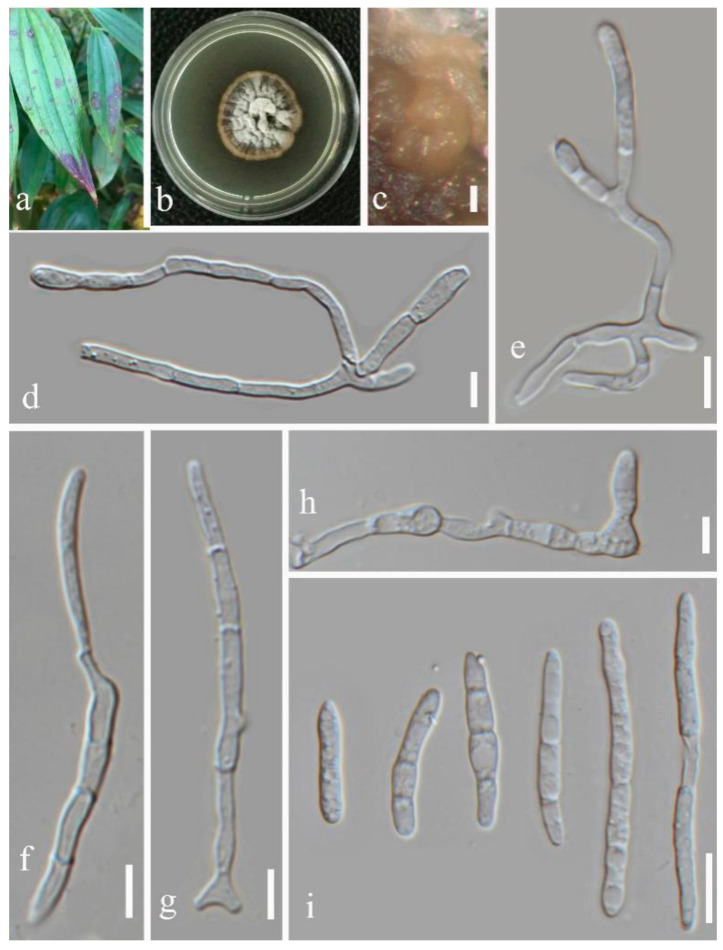
*Septoria dispori* (GUCC 2127.1) (**a**) Leaf spot symptoms on the host. (**b**) Colony on PDA. (**c**) Conidiomata on PDA culture. (**d**,**e**) Conidiophores. (**f**–**h**) Conidiogenous cells and conidia. (**i**) Conidia. Scale bars: (**c**) = 20 µm, (**d**) = 10 µm (**e**) = 5 µm, (**f**–**i**) = 10 µm.

**Figure 7 jof-07-00483-f007:**
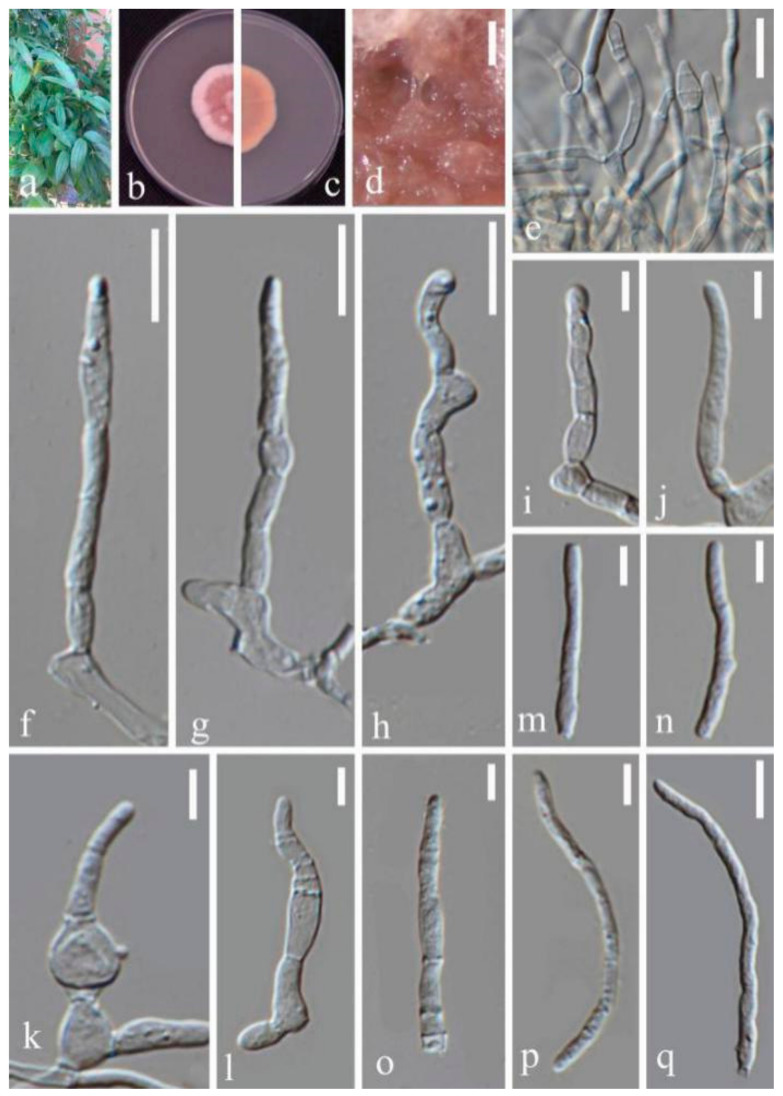
*Septoria protearum* (GUCC 2127.3) (**a.** Leaf spot symptoms on the host. (**b**,**c**) Colony on PDA. (**b**) From above; **c.** from below). (**d**) Mycelium. (**e**) Conidiophores. (**f**–**l**) Conidiogenous cells and conidia. (**m**–**q**) Conidia. Scale bars: (**d**) = 125 µm. (**e**–**h**) = 10 µm. (**i**) = 5 µm. (**j**–**q**) = 10 µm.

**Table 1 jof-07-00483-t001:** Strains numbers and GenBank accession numbers for phylogenetic study.

Species	Isolate No.	GenBank Accession No.				
		*tef1*	*tub2*	*rpb2*	*LSU*	ITS
*Cercospora beticola*	CBS 124.31	KF253246	KF252780	KF252304	KF251802	KF251298
*Septoria aegopodina*	CBS 123741	KF253282	KF252807	–	KF251838	KF251334
*S. anthrisci*	CBS 109020	KF253286	KF252811	KF252340	KF251843	KF251339
*S. anthurii*	CBS 346.58	KF253288	KF252813	KF252342	KF251845	KF251341
*S. apiicola*	CBS 400.54	KF253292	KF252817	KF252346	KF251849	KF251345
*S. astericola*	CBS 128593	KF253294	KF252819	KF252348	KF251851	KF251347
*S. astragali*	CBS 109116	KF253298	KF252823	KF252352	KF251855	KF251351
*S. atropurpurea*	CBS 348.58	KF253299	KF252824	KF252353	KF251856	KF251352
*S. bothriospermi*	CBS 128599	KF253301	KF252826	KF252355	KF251858	KF251354
*S. bupleuricola*	CBS 128603	KF253303	KF252828	KF252357	KF251860	KF251356
*S. calendulae*	CBS 349.58	KF253304	KF252829	KF252358	KF251861	KF251357
*S. callistephi*	CBS 128590	KF253305	KF252830	KF252359	KF251862	KF251358
*S. campanulae*	CBS 128604	KF253308	KF252833	KF252362	KF251865	KF251361
*S. carvi*	KML 1833	–	–	–	–	KX453687
*S. cerastii*	CBS 128612	KF253311	KF252836	KF252365	KF251868	KF251364
*S.* cf. *agrimoniicola*	CBS 128602	KF253284	KF252809	KF252338	KF251841	KF251337
*S.* cf. *rubi*	CBS 128646	KF253314	KF252839	KF252368	KF251871	KF251367
*S.* cf. *sonchi*	CBS 128757	KF253500	KF253020	KF252546	KF252057	KF251552
*S.* cf. *stachydicola*	CBS 128662	KF253513	KF253034	KF252559	KF252071	KF251566
*S. chamaecisti*	CBS 350.58	KF253318	KF252843	KF252372	KF251875	KF251371
*S. chelidonii*	CBS 128607	KF253319	KF252844	KF252373	KF251876	KF251372
*S. chromolaenae*	CBS 113373 ^T^	KF253321	KF252846	KF252375	KF251878	KF251374
*S. chrysanthemella*	CBS 128716	KF253325	KF252850	KF252379	KF251882	KF251378
*S. cirsii*	CBS 128621	KF253328	KF252853	KF252382	KF251885	KF251381
*S. citri*	CBS 315.37	KF253465	–	KF252511	KF252021	KF251516
*S. citricola*	CBS 356.36 ^T^	KF253329	KF252854	KF252383	KF251886	KF251382
*S. clematidis*	CBS 108983	KF253330	KF252855	KF252384	KF251887	KF251383
*S. codonopsidis*	CBS 128620	KF253333	KF252858	KF252387	KF251890	KF251386
*S. convolvuli*	CBS 128627	KF253336	KF252861	KF252390	KF251893	KF251389
*S. coprosmae*	CBS 113391	KF253255	KF252787	KF252313	KF251812	KF251308
*S. crepidis*	CBS 128619	KF253338	KF252863	KF252392	KF251895	KF251391
*S. cretae*	CBS 135095 ^T^	–	KF252720	–	KF251736	KF251233
*S. cruciatae*	CBS 123747	KF253340	KF252865	KF252394	KF251897	KF251393
*S. cucubali*	CBS 102386	KF253344	KF252869	KF252398	KF251901	KF251397
*S. cucurbitacearum*	CBS 178.77	KF253346	–	KF252400	KF251903	KF251399
*S. dearnessii*	CBS 128624	KF253347	KF252871	KF252401	KF251904	KF251400
*S. digitalis*	CBS 391.63	KF253349	KF252873	KF252403	KF251906	KF251402
*S. dispori*	GUCC 2127.1 ^T^	MT996515	MT984348	MT993632	MT985366	MT974584
*S. dispori*	GUCC 2164.3	MT996523	MT984357	MT993641	MT985375	MT974593
*S. dispori*	GUCC 2164.4	MT996524	MT984358	MT993642	MT985376	MT974594
*S. dispori*	GUCC 2127.4	MT996517	MT984350	MT993634	MT985368	MT974586
*S. dolichospora*	CBS 129152	KF253350	KF252874	–	KF251907	KF251403
*S. dysentericae*	CBS 131892	KF253353	KF252877	KF252406	KF251910	KF251406
*S. ekmaniana*	CBS 113612	KF253355	KF252879	–	KF251912	KF251408
*S. epambrosiae*	CBS 128629	KF253356	KF252880	KF252407	KF251913	KF251409
*S. epilobii*	CBS 109084 ^T^	KF253358	KF252882	KF252409	KF251915	KF251411
*S. erigerontis*	CBS 109094	KF253360	KF252884	KF252411	KF251917	KF251413
*S. eucalyptorum*	CBS 118505 ^T^	KF253365	KF252889	KF252415	KF251921	KF251417
*S. exotica*	CBS 163.78	KF253366	KF252890	KF252416	KF251922	KF251418
*S. galeopsidis*	CBS 102411 ^T^	KF253372	KF252896	KF252422	KF251928	KF251424
*S. gentianae*	CBS 128633	KF253374	KF252898	KF252424	KF251930	KF251426
*S. gerberae*	CBS 410.61	KF253468	KF252988	KF252514	KF252024	KF251519
*S. glycines*	CBS 336.53	KF253377	KF252901	–	KF251933	KF251429
*S. glycinicola*	CBS 128618 ^T^	KF253378	KF252902	KF252427	KF251934	KF251430
*S. hederae*	CBS 566.88	KF253470	KF252990	KF252515	KF252026	KF251521
*S. helianthi*	CBS 123.81	KF253379	KF252903	KF252428	KF251935	KF251431
*S. helianthicola*	CBS 122.81	KF253380	KF252904	KF252429	KF251936	KF251432
*S. hibiscicola*	CBS 128615	KF253382	KF252906	KF252431	KF251938	KF251434
*S. hippocastani*	CPC 23103	KF253510	KF253031	KF252556	KF252068	KF251563
*S. justiciae*	CBS 128625	KF253385	KF252909	KF252434	KF251941	KF251437
*S. lactucae*	CBS 108943	KF253387	KF252911	KF252436	KF251943	KF251439
*S. lamiicola*	CBS 123884	KF253397	KF252921	KF252446	KF251953	KF251449
*S. lepidiicola*	CBS 128635	KF253398	KF252922	KF252447	KF251954	KF251450
*S. leptostachyae*	CBS 128613	KF253399	KF252923	KF252448	KF251955	KF251451
*S. leucanthemi*	CBS 109090	KF253403	KF252927	KF252452	KF251959	KF251455
*S. limonum*	CBS 419.51	KF253407	KF252931	KF252456	KF251963	KF251459
*S. linicola*	CBS 316.37	KF253408	KF252932	KF252457	KF251964	KF251460
*S. lobeliae*	CBS 113392	KF253460	KF252981	KF252507	KF252016	KF251511
*S. longipes*	GUCC 2131.1 ^T^	–	MT984351	MT993635	MT985369	MT974587
*S. lycoctoni*	CBS 109089	KF253409	KF252933	KF252458	KF251965	KF251461
*S. lycopersici*	CBS 128654	KF253410	KF252934	KF252459	KF251966	KF251462
*S. lycopicola*	CBS 128651	KF253412	KF252936	KF252461	KF251968	KF251464
*S. lysimachiae*	CBS 102315	KF253413	KF252937	KF252462	KF251969	KF251465
*S. malagutii*	CBS 106.80 ^T^	KF253418	–	KF252467	KF251974	KF251470
*S. matricariae*	CBS 109001	KF253420	KF252943	KF252469	KF251976	KF251472
*S. mazi*	CBS 128755	KF253422	KF252945	KF252471	KF251978	KF251474
*S. melissae*	CBS 109097	KF253423	KF252946	KF252472	KF251979	KF251475
*S. menthae*	CBS 404.34	KF253424	KF252947	–	KF251980	KF251476
*S. napelli*	CBS 109105	KF253426	KF252949	KF252474	KF251982	KF251478
*S. obesa*	CBS 128623	KF253429	KF252952	KF252477	KF251985	KF251481
*S. oenanthicola*	CBS 128649 ^T^	KF253433	KF252954	KF252239	KF251737	KF251234
*S. oenanthis*	CBS 128667	KF253432	KF252953	-	KF251989	KF251485
*S. orchidearum*	CBS 128631 ^T^	KF253434	KF252955	KF252482	KF251990	KF251486
*S. pachyspora*	CBS 128652	KF253437	KF252958	KF252485	KF251993	KF251488
*S. paridis*	CBS 109111	KF253438	KF252959	KF252486	KF251994	KF251489
*S. passifloricola*	CBS 102701	KF253442	KF252963	KF252490	KF251998	KF251493
*S. perillae*	CBS 128655	KF253444	KF252965	KF252491	KF252000	KF251495
*S. petroselini*	CBS 182.44	KF253446	KF252967	KF252493	KF252002	KF251497
*S. phlogis*	CBS 128663	KF253448	KF252969	KF252495	KF252004	KF251499
*S. pileicola*	GUCC 2131.3	MT996519	MT984353	MT993637	MT985371	MT974589
*S. pileicola*	GUCC 2131.4 ^T^	MT996520	MT984354	MT993638	MT985372	MT974590
*S. polygonorum*	CBS 109834	KF253453	KF252974	KF252500	KF252009	KF251504
*S. posoniensis*	CBS 128645	KF253456	KF252977	KF252503	KF252012	KF251507
*S. protearum*	CBS 778.97 ^T^	KF253472	KF252992	KF252517	KF252028	KF251523
*S. protearum*	GUCC 2127.3	MT996516	MT984349	MT993633	MT985367	MT974585
*S. pseudonapelli*	CBS 128664 ^T^	KF253475	KF252995	KF252520	KF252031	KF251526
*S. putrida*	CBS 109088	KF253477	KF252997	KF252522	KF252033	KF251528
*S. rumicum*	CBS 503.76	KF253478	KF252998	KF252523	KF252034	KF251529
*S. saccardoi*	CBS 128756	KF253479	KF252999	KF252524	KF252035	KF251530
*S. sanguisorbigena*	GUCC 2131.2	MT996518	MT984352	MT993636	MT985370	MT974588
*S. sanguisorbigena*	GUCC 2164.1	MT996521	MT984355	MT993639	MT985373	MT974591
*S. sanguisorbigena*	GUCC 2164.2 ^T^	MT996522	MT984356	MT993640	MT985374	MT974592
*S. scabiosicola*	CBS 109093	KF253487	KF253007	KF252532	KF252043	KF251538
*S. senecionis*	CBS 102366 ^T^	KF253492	KF253012	KF252538	KF252049	KF251544
*S. siegesbeckiae*	CBS 128659	KF253494	KF253014	KF252540	KF252051	KF251546
*S. sii*	CBS 102370	KF253497	KF253017	KF252543	KF252054	KF251549
*S. sisyrinchii*	CBS 112096	KF253499	KF253019	KF252545	KF252056	KF251551
*S. stachydicola*	CBS 128668	KF253512	KF253033	KF252558	KF252070	KF251565
*S. stachydis*	CBS 109127	KF253517	KF253038	KF252563	KF252075	KF251570
*S. stellariae*	CBS 102376	KF253521	KF253042	KF252567	KF252079	KF251574
*S. taraxaci*	CBS 567.75	KF253524	KF253045	KF252570	KF252082	KF251577
*S. tinctoriae*	CBS 129154	KF253525	KF253046	KF252571	KF252083	KF251578
*S. tormentillae*	CBS 128647	KF253527	KF253048	KF252573	KF252085	KF251580
*S. urticae*	CBS 102375 ^T^	KF253530	KF253051	KF252576	KF252088	KF251583
*S. verbascicola*	CBS 102401	KF253531	KF253052	KF252577	KF252089	KF251584
*S. verbenae*	CBS 113438	KF253532	KF253053	KF252578	KF252090	KF251585
*S. villarsiae*	CBS 514.78	KF253534	KF253055	KF252580	KF252092	KF251587
*S. violae-palustris*	CBS 128644	KF253537	KF253058	KF252583	KF252095	KF251590

Ex-type isolates are labeled with “^T^”.

**Table 2 jof-07-00483-t002:** Primers, primer sequences, and thermal cycling program for PCR amplification.

Locus	Primer	Primer Sequence 5′ to 3′	Annealing Temperature (°C)	Direction	Reference
*tef1*	EF1-728F	CATCGAGAAGTTCGAGAAGG	52	Forward	[18]
	EF-2	GGARGTACCAGTSATCATGTT		Reverse	[19]
*tub2*	T1	AACATGCGTGAGATTGTAAGT	52	Forward	[20]
	β-Sandy-R	GCRCGNGGVACRTACTTGTT		Reverse	[21]
*rpb2*	fRPB2-5F	GAYGAYMGWGATCAYTTYGG	49	Forward	[22]
	fRPB2-414R	ACMANNCCCCARTGNGWRTTRTG		Reverse	[23]
*LSU*	LSU1Fd	GRATCAGGTAGGRATACCCG	52	Forward	[24]
	LR5	TCCTGAGGGAAACTTCG		Reverse	[25]
ITS	ITS5	GGAAGTAAAAGTCGTAACAAGG	52	Forward	[26]
	ITS4	TCCTCCGCTTATTGATATGC		Reverse	[26]

**Table 3 jof-07-00483-t003:** Parameters for MP analyses.

	Total Characters	Number of Parsimony-Informative Characters	TL	CI	RI	HI	RC
ITS	486	43	176	0.642	0.76	0.358	0.488
LSU	799	31	112	0.625	0.863	0.375	0.539
*rpb2*	345	18	780	0.273	0.746	0.727	0.204
*tef1*	469	231	1498	0.379	0.707	0.621	0.268
*tub2*	325	165	1221	0.326	0.774	0.674	0.252
*tef1* + *rpb2* + *tub2* + ITS	1625	548	3927	0.328	0.716	0.672	0.235
*tef1* + *rpb2* + *tub2* + ITS + *LSU*	2434	567	4075	0.330	0.720	0.670	0.238

## Data Availability

The data presented in this study are available in Appendix A.

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
