# Peer review of "Molecular and Morphological Assessment of Septoria Species Associated with Ornamental Plants in Yunnan Province, China"

_jof, 2021, doi:10.3390/jof7060483_

Round 1
Reviewer 1 Report
1. The authors present a scientifically sound manuscript that sheds some light on variability of the Septoria species concept. Although not especially enthusiastic on describing new species with often only few genetic differences, the reviewer is essentially convinced by the data that the authors are on a reasonable track. Morphologically as well as at the sequence level the arguments are convincing. The authors should, however, make clear their ideas about the species concept that they apply.
2. The reviewer strengthens the argumment that LSU sequence data, at least in this case, are not the best way for identification and certainly not for describing new species. The alternative and rather conserved genes are much more suitable. But how, in this line of argumantation, do the ITS data fit in -and especially why are they not also excluded? This point needs much more attention and certainly more profound discussion. Maybe all rDNS sequences describe more the history of the nucleolus equivalent than that of the nucleus? What do these sequences mean, if intranuclear variations are taken into account? Are there data on rDNA copy numbers and especially variability?
3. Figures 1, 2: Look for maximum contrast in print, which is always black on white. Thus, please remove the yellow background colour, and maybe replace the green background simply by framing these regions or similar.
4. The text is well-written throughout and needs only very few adjustments.
The reviewer would be glad to see this manuscript in print after carefully considering the suggestions.
Author Response
Dear Reviewer 1
Thank you very much for your help to improve our manuscript. Now we have finished the revision (attachment) and hope to re-submit it. We would like to reply your comments point by point.
1. The authors present a scientifically sound manuscript that sheds some light on variability of the Septoria species concept. Although not especially enthusiastic on describing new species with often only few genetic differences, the reviewer is essentially convinced by the data that the authors are on a reasonable track. Morphologically as well as at the sequence level the arguments are convincing. The authors should, however, make clear their ideas about the species concept that they apply.
In the Discussion section of revised copy, we supplemented some contents about the Septoria species concept, which was “Quaedvlieg et al. [10] treated species in Septoria within a modern taxonomic framework and pointed out that Septoria spp. formed a well-defined phylogenetic clade. Regarding morphology, the species concept was to produce pycnidial, ostiolate conidiomata; conidiophores reduced to conidiogenous cells that proliferate sympodially; and hyaline, filiform conidia with transverse eusepta that fit the original concept of [4]”.
2. The reviewer strengthens the argumment that LSU sequence data, at least in this case, are not the best way for identification and certainly not for describing new species. The alternative and rather conserved genes are much more suitable. But how, in this line of argumantation, do the ITS data fit in -and especially why are they not also excluded? This point needs much more attention and certainly more profound discussion. Maybe all rDNS sequences describe more the history of the nucleolus equivalent than that of the nucleus? What do these sequences mean, if intranuclear variations are taken into account? Are there data on rDNA copy numbers and especially variability?
In the Discussion section of revised copy, we supplemented some contents to explain why we suggested to exclude LSU but keep ITS sequence data, that was “...whereas the conserved LSU sequences included the least parsimonious characters (31/799) (Table 3). In morphology, all species in Branch 3 produced filiform or fusiform, sub-straight to slightly curved conidia mainly with 3 septa, which was not a unique characteristic. Thus, we proposed to excluding the LSU region for multi-gene analyses of Septoria at the species level, but always as the primary DNA barcode with more parsimonious characters (43/486), the ITS fragment was conserved in the present phylogenetic analysis”.
3. Figures 1, 2: Look for maximum contrast in print, which is always black on white. Thus, please remove the yellow background colour, and maybe replace the green background simply by framing these regions or similar.
In the revised copy, we have removed the yellow background colour, and replaced the green background simply by framing these regions
4. The text is well-written throughout and needs only very few adjustments.
We have used the editing services listed at https://www.mdpi.com/authors/english to improve the language.
With best regards
Yuan-Yan An
Yong Wang
Reviewer 2 Report
The article provides a substantial contribute to the taxonomy of Septoria and four new species of this genus are described. The formal description is correct and phylogenetic analysis of different genetic loci give enough support to the separation of species.
Only minor revisions are requested (see notes in the text, attached file).

Author Response
Dear Reviewer 2
Thank you very much for your help to improve our manuscript. Now we have finished the revision according to your comment. We have also used the editing services listed at https://www.mdpi.com/authors/english to improve the language.
Please see the revised manuscript.
With best regards
Yuan-Yan An
Yong Wang
This manuscript is a resubmission of an earlier submission. The following is a list of the peer review reports and author responses from that submission.